# Nurse's spiritual care competence in Ethiopia: A multicenter cross-sectional study

**Kalid Seid**[ORCID]*, **Adem Abdo**

Department of Nursing, College of Medicine and Health Sciences, Mizan-Tepi University, Mizan, Southwest People Regional State, Ethiopia

* kalidseid7@gmail.com

## Abstract

### Background

Many health care professionals emphasize that spirituality is an important factor in overall health. Although spiritual practices are vital to health, spirituality has received little emphasis in nursing. Hence, the study's purpose has been to evaluate the current state of spiritual care competence and the factors that influence it among nurses in Southwest Ethiopia.

### Methods

From July 1 to 20, 2021, nurses at five hospitals in southwest Ethiopia were enrolled in a facility-based cross-sectional study. The study subjects were chosen using a systematic random sampling. A self-administered questionnaire was undertaken to gather the data. Epi Data 3.1 was used to code the dataset, and SPSS version 25 was used for analysis. To identify factors associated with spiritual care competence, researchers performed bivariate and multivariable linear regression analyses. The significance level was set at $p < 0.05$.

### Results

Three hundred sixty-seven nurses attended in the study, giving a 91.06, percent rate of response. The mean spiritual care competence score among healthcare professionals was $3.14 \pm 0.74$. Age ($p < 0.05$), and training in spiritual care ($p < 0.05$) were significantly associated with spiritual care competence.

### Conclusions

Spiritual care competence was moderate among the nurses. Spiritual care competence varies in accordance with a number of factors, including age, and training in spiritual care. Nurses are better suited to focus on the spiritual health of clients, which necessitates the provision of spiritual care competence training for nurses.

**Data Availability Statement:** All the data underlying this study are provided in the Supporting information file.

**Funding:** The author(s) received no specific funding for this work.

**Competing interests:** The authors have declared that no competing interests exist.

## Introduction

Physical, social, cultural, emotional, and spiritual elements play a role in one's health [1]. Spirituality, on the other hand, is the least recognized and contentious, and the dispute over the meaning and conception of spiritual practice and religiosity persists [2]. Independent of its meaning or conceptualization, spirituality has been found to contribute to people's health and well-being by enhancing physical, cognitive, and social dimensions [3, 4].

Spiritual practices can help patients with chronic diseases to improve their quality of life and cope [5, 6]. People discovered that just by drawing stability and encouragement from spirituality and religious spiritual practices, chronic disease lost its "seriousness" and it was easier to manage and deal with in the context of everyday life [7]. Spiritual development plays a critical role in tackling life barriers and challenges, strengthening the client's perseverance and thereby improving quality of life [8].

Spiritual care assists people, notably in difficult conditions, by boosting spiritual issues [9]. Spiritual care tries to overcome clients' fears, worries, and suffering in an attempt to lessen stress, give hope, and inspire clients to achieve an inner calm [10]. When delivering spiritual care, nurses' beliefs in spirituality might influence their behavior and interactions with patients [11].

Paying attention to clients' worries and fears; working to develop a deep understanding of different faiths, belief systems, and religions; offering solace; realizing the importance of spiritual issues in severely ill and critical clients; as well as the procedures for referring patients to religious leaders or other spiritual counseling services are all cases of spiritual care [12]. Clients' well-being seems to be impacted by unfulfilled emotional and social needs. Some of the awful consequences include poorer quality of life, a high probability of despair, and a decline in cognitive health [13].

Patients may experience better survival, health, quality of life, and optimism as a result of spiritual care provided by health professionals, as well as less dread of loss, solitude, sadness, and loss of meaning [14–16]. In recent years, spirituality has been highlighted as a crucial but frequently overlooked component of patient health [17]. As a result, honoring and responding to clients' spiritual needs, whenever they desire or demand it, should indeed be regarded as a core duty for nurses, not just an "extra" task [18].

Nurses might refuse to provide spiritual support for a variety of reasons, including the perception that a person's faith is just a personal affair, sentiments of merely not having time, difficulties in meeting the client's preferences, and a dread of preaching [19]. To appreciate the spiritual needs of others, nurses must acquire awareness of their own spiritual practices and ideologies [20].

Data on spiritual care competency among Ethiopian nurses, as well as in the research field, are limited. This is Ethiopia's first investigation to evaluate nurses' spiritual care competency. Hence, the study's purpose has been to evaluate the current state of spiritual care competency of nurses in Southwest Ethiopia, and as well as the factors influencing it.

## Materials and methods

### Study design, setting, and population

An institution-based multi-center cross-sectional study was conducted at Mizan Tepi University Teaching Hospital, Agarro general hospital, Gebretsadik shawo general hospital, Shenen gibe general hospital and Jimma medical center, Ethiopia, from July 1 to 20, 2021. The Benchi-Maji zone is home to Mizan Tepi University Teaching Hospital. It is 561 kilometers from Addis Ababa and 844 kilometers from Hawasa. The Mizan-Tepi University Teaching Hospital

is expected to serve more than 829,000 people. In the kefa zone, Gebretsadik Shawo general hospital is located in Bonga town, 464 kilometers from Addis Abeba. Jimma is home to the Jimma Medical Center and the Shenen Gibe General Hospital. They were discovered 355 kilometers from Addis Ababa. Agarro general hospital is located in Jimma woreda.

Nurses working in Mizan Tepi University Teaching Hospital, Agarro General Hospital, Gebresadik Shawo General Hospital, Shenen Gibe General Hospital and Jimma Medical Center were the study's source population. The study included all nurses working at selected public hospitals, as well as those with more than six months of experience.

## Sample size and sampling procedure

The representative sample was determined using a single-population proportion formula. The following parameters have been used to determine the sample size: The proportion of spiritual care competence was 50%, the margin of error was 0.05, the confidence level was 95%, and the rate of non-response was 5%, resulting in a final sample of 403. The number and list of nurses were obtained from each hospital human resources office. Based on this information, the study population was assigned proportionally to each institution. As a result, participants in this study were chosen using systematic sampling. Every 2 nurses were recruited from each of the hospitals. The first research participant was chosen randomly.

## Data collection tools and procedures

The data is taken using a self-administered questionnaires. "The Spiritual Care Competence Scale (SCCS) was used to gather data on nurses' competence in spiritual care. It includes 27 items and six subscales. Each item was rated on a five-point Likert scale ranging from completely disagree to completely agree" [21]. Section two encompasses participants' demographic information, including sexual identity, religion, marital status, and educational status. Additionally, work related factors included clinical experience, the type of ward, organizational position, employment status, and training in spiritual care. Data were gathered by five BSc nurses and two Adult health nurse specialists who served as supervisors.

## Operational definition

**Competence of spiritual care.** "Nurses' competence in spiritual care was measured using 27 items on a 5-point Likert scale with value ranging from 27 to 135. The higher the value, the higher the competence of spirituality and spiritual care" [21]. "The overall mean score was divided by 27, and the nurses had a value ranging from 1 to 5. Then the competence of the spiritual care level was divided into low, moderate and high based on this score. A low level is a mean score between 1 and 2.33, a moderate level is a mean score between 2.34 and 3.67, and a high level is a mean score between 3.68 and 5" [22].

## Data processing and analysis

Epi Data 3.1 was used to code the dataset, and SPSS version 25 was used for analysis. To summarize the data, the frequency, percentage, root-mean square deviation, and mean have all been used as descriptive statistics. The link between spiritual care competency and explanatory variables was first investigated using bivariate linear regression. Variables with $p < 0.25$ in bivariate linear regression were candidates for multiple linear regression. To account for potential confounders, a multiple regressions analysis has been used. The statistical significance level was set at $P < 0.05$. The assumptions of multiple linear regression were tested prior to analyzing the results. The Kolmogorov-Smirnov test validated the normality assumption.

The variance inflation factor (VIF) was used to test the collinearity assumption and determine the correlation between the independent variables. According to the findings, all variables had a VIF of less than 5.

### Ethics approval and consent to participate

Mizan-Tepi University College of Medicine, and Health Sciences Ethics Committee approved the study immediately prior to the start of the investigation. The administrative and unit chiefs of all selected hospitals were also consulted. To maintain confidentiality, names and other private labels were removed from the sheets and reports. The scope of research, benefits of research endeavor, and freedom to leave at any moment were all explained to the respondents. Everyone who took part signed a written consent form. All approaches were carried out in compliance with manuscript standards and regulations.

## Results

### Socio-demographic characteristics

Of the 403 invited participants, 367 completed the questionnaires, yielding a 91.06, percent response rate. The participants' average age was 22.69 years (SD = ±12.59 years), and the majority (41.1%) were in the 25–29-year age group. Two hundred twenty-one (60.2%) were male and 160(43.6%) were Orthodox. More than two-thirds (68.4%) of them held Bachelor of Science (B.Sc.) degrees through nursing (**Table 1**).

### Work related factors

The mean clinical experience of the respondents was 4.87 (SD ±3.78), of which 1/3rd (33.2%) had 5 to 9 years of clinical experience. The majority (87.5%) were staff nurses in the current organizational position. Three hundred and sixty-two (98.6%) were formal employment types. The majority of participants (87.2%) did not receive any spiritual or religious training (**Table 2**).

**Table 1. Nurses' socio-demographic characteristics at selected public hospitals in southwest Ethiopia in 2021 (n = 367).**

| Variables | Category | Frequency | Percentage |
|---|---|---|---|
| Age | <25 | 88 | 24.0 |
| M 22.69 | 25–29 | 151 | 41.1 |
| SD ±12.59 | 30–34 | 81 | 22.1 |
| | > = 35 | 47 | 12.8 |
| Sex | Male | 221 | 60.2 |
| | Female | 146 | 39.8 |
| Religion | Orthodox | 160 | 43.6 |
| | Muslim | 142 | 38.7 |
| | Protestant | 61 | 16.6 |
| | Other | 4 | 1.1 |
| Marital status | Single | 175 | 47.7 |
| | Married | 175 | 47.7 |
| | Divorced | 17 | 4.6 |
| Educational status | Diploma | 90 | 24.5 |
| | Bachelor degree | 251 | 68.4 |
| | Master's degree | 26 | 7.1 |

**Table 2. Nurses' work-related characteristics in selected public hospitals in southwest Ethiopia in 2021 (n = 367).**

| Variables | Category | Frequency | Percentage |
|---|---|---|---|
| **Clinical experience in year** | <2 | 108 | 29.4 |
| **Mean 4.87** | 2–4 | 95 | 25.9 |
| | 5–9 | 122 | 33.2 |
| **SD ±3.78** | 10–14 | 27 | 7.4 |
| | > = 15 | 15 | 4.1 |
| **Current organizational position** | Staff nurse | 321 | 87.5 |
| | Head nurse | 39 | 10.6 |
| | Supervisor nurse | **7** | 1.9 |
| **Employment type** | Formal | 362 | 98.6 |
| | Contractual | 5 | 1.4 |
| **Types of wards** | Medical | 62 | 16.9 |
| | Surgical | 46 | 12.5 |
| | Pediatric | 61 | 16.6 |
| | Emergency | 64 | 17.4 |
| | Outpatient | 58 | 15.8 |
| | ICU | 47 | 12.8 |
| | Burn unit | 15 | 4.1 |
| | Oncology | 14 | 3.8 |
| **Training on spiritual care** | Yes | 47 | 12.8 |
| | No | 320 | 87.2 |

## The level of spiritual care competence

The mean and standard deviation of the spiritual care competence level were calculated. The average SCCS result was 3.14 (SD = ±0.74) out of a possible total of 5 points, indicating a moderate level of spiritual care competence. The mean spiritual care competence scores for knowledge of "Assessment and implementation of spiritual care", "Professionalization and improving the quality of spiritual care", "Personal support and patient counseling", "Referral", "Attitude towards patient spirituality and Communication" were 3.24(SD±0.8), 3.04(SD±0.86), 3.03(SD±0.85), 3.09 (SD±0.09), 3.13(SD±0.96), and 3.29(SD±1.20) respectively. The lowest mean score, 3.03(SD ±0.85), was found for knowledge of "personal support and patient counseling". The study's findings show that participants rated their spiritual care competency as moderate (**Table 3**).

## Factors associated with spiritual care competence

Multivariable linear regression analyses revealed factors associated with spiritual competency among nurses. In a bivariate linear regression, age, marital status, clinical experience in years

**Table 3. The spiritual care competence of nurses at selected public hospitals in southwest Ethiopia in 2021 (n = 367).**

| Variable | Possible scores | Mean (standard deviation) | Minimum | Maximum |
|---|---|---|---|---|
| "Assessment and implementation of spiritual care scale" | 1–5 | 3.24(SD±0.8) | 1.00 | 5.00 |
| "Professionalization and improving the quality of spiritual care scale" | 1–5 | 3.04(SD±0.86) | 1.00 | 5.00 |
| "Personal support and patient counseling scale | 1–5 | 3.03(SD±0.85) | 1.00 | 5.00 |
| Referral scale" | 1–5 | 3.09(SD±0.09) | 1.00 | 5.00 |
| "Attitude towards patient spirituality scale" | 1–5 | 3.13(SD±0.96) | 1.00 | 5.00 |
| "Communication scale" | 1–5 | 3.29(SD±1.20) | 1.00 | 5.00 |
| Total SCCS scale score | 1–5 | 3.14(SD±0.74) | 1.00 | 5.00 |

and training on spiritual care were found to be substantially associated with spiritual care competency among nurses at p<0.25. To investigate factors related to spiritual care competency, independent variables with p<0.25 in the bivariate linear regression analysis were added to the multivariable linear regression analysis. At a significance level of 0.05, the backward elimination approach was used to choose the variables for the final model.

The findings revealed that age, and spiritual care training were significantly associated with spiritual care competencies among nurses. Accordingly, a one-unit increase in age resulted in a -0.026 unit decrease in spiritual care competence (β = -0.026, p = 0.001). Training in spiritual care increased spiritual care competence by 0.238 times compared to those who didn't receive any training in spiritual care (β = 0.238, p = 0.039) (**Table 4**).

## Discussion

This study assessed the current state of spiritual care competence and its associated factors among nurses. According to the findings, the mean score for a nurse's spiritual care competence was 3.14. This indicates that there is a moderate level of spiritual care competence among nurses in southwest Ethiopia.

These findings were better than those of a study done in Pakistan, where the mean score was 2.5 [23]. The disparity could be attributed to a difference in sample size, as the previous study only included nurses working in the Corona Virus Disease (COVID) unit. Compared to the results of a study conducted in Slovakia, which had a mean score of 3.72 [24], the current findings were lower. This disparity could be attributed to differences in socio-demographic features, sample sizes, and research settings.

Another notable element of this research was the identification of spiritual competence-related characteristics. As a corollary, age, and training in spiritual care were found to be related to spiritual care competency.

The current study found that older nurses had lower spiritual care competence scores. This finding is supported by a Saudi Arabian study [25], which found that being in the 40–49 age range reduced spiritual care competence. The current study's findings also showed a significant difference between nurses' scores on spiritual care competence and those receiving spiritual care training. This conclusion is supported by studies conducted in Iran [26, 27], that found a significant difference between nurses' spiritual care competence and spiritual issue training. Spirituality education enables learners to develop a significant sense of spiritual knowledge, broaden their views on spiritual care, and strengthen their skills in identifying and addressing clients' desires [28, 29].

**Table 4. Results of multivariable linear regression among nurses at selected public hospitals in southwest Ethiopia in 2021.**

| Predictor variable | Unstandardized coefficient | | p-value | 95% CI | |
|---|---|---|---|---|---|
| | B | SE | | Upper | Lower |
| **Age (in year)** | -0.026 | 0.008 | **0.001** | 0.041 | 0.011 |
| **Marital status** | | | | | |
| Single(reference) | | | | | |
| Married | -0.021 | 0.087 | 0.808 | -0.191 | 0.149 |
| Divorced | 0.024 | 0.87 | 0.298 | -0.564 | 0.173 |
| **Clinical experience in year** | 0.007 | 0.020 | 0.736 | -0.033 | 0.047 |
| **Training in Spiritual care** | | | | | |
| Yes | 0.238 | 0.115 | **0.039** | 0.012 | 0.464 |
| No(reference) | | | | | |

The findings of this study demonstrated no significant relation between marital status, educational level, clinical experience, and spiritual care competency. This conclusion is supported by studies from Iran [26] and Malaysia [30], which revealed no link between spiritual care competency and socio-demographic characteristics.

This research has certain limitations. The study used self-reported data from nurses, which could have resulted in social desirability bias. The results may not be generalizable to other Ethiopian hospitals and medical centers because the study was limited to five hospitals in southwest Ethiopia. Because the research was cross-sectional, causality could not be determined.

## Conclusions

Nurses' spiritual care competence was moderate. These findings on spiritual care competency differ depending on age and training in spiritual care. The findings of this research encourage Ethiopian nursing curriculum developers to include spiritual care competence in their current curricula, as well as the necessity for spiritual care competency training for nurses. Future researchers, including nurses working in private clinics, should conduct further research. Longitudinal study with a large sample size is required to determine cause-and-effect relationships.

## Supporting information

**S1 File. Data collection tool.**
(DOCX)

## Acknowledgments

The authors are grateful to Mizan-Tepi University for permission to undertake this study. We'd like to express our heartfelt appreciation to the hospitals and personnel chosen for their ongoing assistance. Ultimately, we would like to apply data collectors to all the participants in the study.

## Author Contributions

**Conceptualization:** Kalid Seid.

**Data curation:** Kalid Seid, Adem Abdo.

**Formal analysis:** Kalid Seid, Adem Abdo.

**Funding acquisition:** Kalid Seid, Adem Abdo.

**Investigation:** Kalid Seid, Adem Abdo.

**Methodology:** Kalid Seid, Adem Abdo.

**Project administration:** Kalid Seid, Adem Abdo.

**Resources:** Kalid Seid, Adem Abdo.

**Software:** Kalid Seid, Adem Abdo.

**Supervision:** Kalid Seid.

**Validation:** Kalid Seid, Adem Abdo.

**Visualization:** Kalid Seid, Adem Abdo.

**Writing – original draft:** Kalid Seid, Adem Abdo.

**Writing – review & editing:** Kalid Seid, Adem Abdo.

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
