## [Decision Letter · Decision Letter 0]

13 Jan 2022

PONE-D-21-35932Nurse’s Spiritual Care Competence in Ethiopia: a multicenter cross-sectional studyPLOS ONE

Thank you for submitting your manuscript to PLOS ONE. After careful consideration, we feel that it has merit but does not fully meet PLOS ONE’s publication criteria as it currently stands. Therefore, we invite you to submit a revised version of the manuscript that addresses the points raised during the review process.

We look forward to receiving your revised manuscript.

Kind regards,

Luigi Lavorgna

Academic Editor

PLOS ONE

Journal Requirements:

a) Did participants provide their written or verbal informed consent to participate in this study?

3.Please review your reference list to ensure that it is complete and correct. If you have cited papers that have been retracted, please include the rationale for doing so in the manuscript text, or remove these references and replace them with relevant current references. Any changes to the reference list should be mentioned in the rebuttal letter that accompanies your revised manuscript. If you need to cite a retracted article, indicate the article’s retracted status in the References list and also include a citation and full reference for the retraction notice.

4. Thank you for submitting the above manuscript to PLOS ONE. During our internal evaluation of the manuscript, we found significant text overlap between your submission and the following previously published works, some of which you are an author.

-https://journals.plos.org/plosone/article?id=10.1371%2Fjournal.pone.0254643

-https://journals.sagepub.com/doi/10.1177/0969733015600910

-https://onlinelibrary.wiley.com/doi/10.1111/ppc.12651

-https://onlinelibrary.wiley.com/doi/10.1111/inr.12222

-https://eprints.arums.ac.ir/11483/1/Spiritual%20perspectives.%20APJON.pdf

Please revise the manuscript to rephrase the duplicated text, cite your sources, and provide details as to how the current manuscript advances on previous work. Please note that further consideration is dependent on the submission of a manuscript that addresses these concerns about the overlap in text with published work.

Reviewers' comments:

Reviewer's Responses to Questions

**Comments to the Author**

1. Is the manuscript technically sound, and do the data support the conclusions?

Reviewer #1: Yes

2. Has the statistical analysis been performed appropriately and rigorously? 

Reviewer #1: Yes

3. Have the authors made all data underlying the findings in their manuscript fully available?

Reviewer #1: Yes

4. Is the manuscript presented in an intelligible fashion and written in standard English?

Reviewer #1: Yes

5. Review Comments to the Author

Reviewer #1: In this study, Seid and Abdo assessed the current status of competence in spiritual care and its associated factors among nurses in Southwest Ethiopia. The article is clear, and methods are sound.

Just two concerns:

Spirituality has been identified as an essential factor in overall health and wellbeing; however, it appears essential, especially in coping with chronic illness. Therefore, in the introduction section, concerning the concept that spirituality contributes to health and wellbeing, the authors should briefly discuss the impact of religiosity and spirituality on quality of life, especially in chronic diseases (suggested references: PMID: 34816315; PMID: 22083464).

A further concern: in which language the questionnaire was administered. If not in English, is the questionnaire validated in the language used?

6. PLOS authors have the option to publish the peer review history of their article (what does this mean?). If published, this will include your full peer review and any attached files.

Reviewer #1: No

---

## [Author Response · Author response to Decision Letter 0]

11 Feb 2022

Author’s response to reviews

Title: Nurse’s spiritual care competence in Ethiopia: a multicenter cross-sectional study

Authors:

Kalid Seid (Kalidseid7@gmail.com)

Adem Abdo (ademabdo448@gmail.com)

Version: 1 Date: 11 February 2022

Author’s response to reviews:

General response:

Thank you for giving us the opportunity to submit a revised draft of our manuscript titled “Nurse’s spiritual care competence in Ethiopia: a multicenter cross-sectional study” to [PLOS ONE]. We appreciate the time and effort that editors and reviewers have dedicated to providing your valuable feedback on our manuscript. We are grateful to the editors and reviewers for their insightful comments on our paper. We have been able to incorporate changes to reflect most of the suggestions provided by the editors and reviewers. We hope that the revised manuscript has now addresses all of the editors and reviewers comments. Please find attached the point-by-point response to those comments as well as the revised manuscript with track changes as well as the clean copy manuscript. In preparing the manuscript, we have strictly followed journal instructions for authors. Looking forward to hearing from you soon.

Here is a point-by-point response to the editor’s comments and concerns.

Comments from Editor

Comment 1: [When submitting your revision, we need you to address these additional requirements. Please ensure that your manuscript meets PLOS ONE's style requirements, including those for file naming. The PLOS ONE style templates can be found at 

https://journals.plos.org/plosone/s/file?id=wjVg/PLOSOne_formatting_sample_main_body.pdf andhttps://journals.plos.org/plosone/s/file?id=ba62/PLOSOne_formatting_sample_title_authors_affiliations.pdf]

Response: Thank you for pointing this out. During revision we thoroughly follow the journal style requirements. We incorporated the change in track changes as well as clean copy of the revised manuscript.

Comment 2: [Please amend your current ethics statement to address the following concerns:

a) Did participants provide their written or verbal informed consent to participate in this study?

b) If consent was verbal, please explain i) why written consent was not obtained, ii) how you documented participant consent, and iii) whether the ethics committees/IRB approved this consent procedure.”

Response: Thank you for all of your comments and all those concerns are appreciated. We incorporated in revised version as “Mizan-Tepi University College of Medicine, and Health Sciences Ethics Committee approved the study immediately prior to the start of the investigation”. Regarding the “consent”, we provided written informed consent for all participants and all approaches were carried out in compliance with manuscript standards and regulations. We highlighted the change within the track change and you can also see from clean copy of revised version in Ethical approval and consent to participate statement.

Comment 3: [Please review your reference list to ensure that it is complete and correct. If you have cited papers that have been retracted, please include the rationale for doing so in the manuscript text, or remove these references and replace them with relevant current references. Any changes to the reference list should be mentioned in the rebuttal letter that accompanies your revised manuscript. If you need to cite a retracted article, indicate the article’s retracted status in the References list and also include a citation and full reference for the retraction notice.]

Response: Thank you for your valuable feedback. We have thoroughly checked all references and we didn’t get retracted paper. We added reference number 5, 6, and 7 to address the impact of spirituality on chronic illness patients that raised by the reviewer. We also added reference number 21 to address operational definition in the manuscript. Finally, we edit reference number 23 since it lacks journal name, year, volume, and issue. We highlighted the change within the track change and you can also see from clean copy of revised version.

Comment 4: [Thank you for submitting the above manuscript to PLOS ONE. During our internal evaluation of the manuscript, we found significant text overlap between your submission and the following previously published works, some of which you are an author.

-https://journals.plos.org/plosone/article?id=10.1371%2Fjournal.pone.0254643

-https://journals.sagepub.com/doi/10.1177/0969733015600910

-https://onlinelibrary.wiley.com/doi/10.1111/ppc.12651

-https://onlinelibrary.wiley.com/doi/10.1111/inr.12222

-https://eprints.arums.ac.ir/11483/1/Spiritual%20perspectives.%20APJON.pdf

Please revise the manuscript to rephrase the duplicated text, cite your sources, and provide details as to how the current manuscript advances on previous work. Please note that further consideration is dependent on the submission of a manuscript that addresses these concerns about the overlap in text with published work.

We will carefully review your manuscript upon resubmission, so please ensure that your revision is thorough.]

Response: Thank you for your valuable feedback. All overlapping texts, as well as spelling and grammatical errors pointed out have been corrected in the revised version of the manuscript. 

Here is a point-by-point response to the reviewer comments and concerns.

Comments from Reviewer 1

Comment 1: [In this study, Seid and Abdo assessed the current status of competence in spiritual care and its associated factors among nurses in Southwest Ethiopia. The article is clear, and methods are sound.]

Response: Thank you for the kind gestures! I appreciate you and all that you do.

Comment 2: [Spirituality has been identified as an essential factor in overall health and wellbeing; however, it appears essential, especially in coping with chronic illness. Therefore, in the introduction section, concerning the concept that spirituality contributes to health and wellbeing, the authors should briefly discuss the impact of religiosity and spirituality on quality of life, especially in chronic diseases (suggested references: PMID: 34816315; PMID: 22083464).”

Response: Thank you for all of your comments and all those concerns are appreciated. We incorporated the change in Introduction section paragraph 2. We highlighted the change within the track change and you can also see from clean copy of revised version.

Comment 3: [A further concern: in which language the questionnaire was administered. If not in English, is the questionnaire validated in the language used?]

Response: Thank you for raising important issue. Since all nurses have diploma and above academic background, we administered the adopted English Version questionnaire.

---

## [Editor Report · Decision Letter 1]

21 Feb 2022

PONE-D-21-35932R1Nurse’s spiritual care competence in Ethiopia: a multicenter cross-sectional studyPLOS ONE

Thank you for submitting your manuscript to PLOS ONE. After careful consideration, we feel that it has merit but does not fully meet PLOS ONE’s publication criteria as it currently stands. Therefore, we invite you to submit a revised version of the manuscript that addresses the points raised during the review process.

We look forward to receiving your revised manuscript.

Kind regards,

Luigi Lavorgna

Academic Editor

PLOS ONE

Journal Requirements:

Additional Editor Comments:

Add any references that the reviewer has indicated

---

## [Author Response · Author response to Decision Letter 1]

22 Feb 2022

Author’s response to reviews

Title: Nurse’s spiritual care competence in Ethiopia: a multicenter cross-sectional study

Authors:

Kalid Seid (Kalidseid7@gmail.com)

Adem Abdo (ademabdo448@gmail.com)

Version: 2 Date: 22 February 2022

Author’s response to reviews:

General response:

Thank you for giving us the opportunity to submit a revised draft of our manuscript titled “Nurse’s spiritual care competence in Ethiopia: a multicenter cross-sectional study” to [PLOS ONE]. We appreciate the time and effort that editors and reviewers have dedicated to providing your valuable feedback on our manuscript. We are grateful to the editors and reviewers for their insightful comments on our paper. We have been able to incorporate changes to reflect most of the suggestions provided by the editors and reviewers. We hope that the revised manuscript has now addresses all of the editors and reviewers comments. Please find attached the point-by-point response to those comments as well as the revised manuscript with track changes as well as the clean copy manuscript. In preparing the manuscript, we have strictly followed journal instructions for authors. Looking forward to hearing from you soon.

Here is a point-by-point response to the editor’s comments and concerns.

Comments from Editor

Comment 1: [Please review your reference list to ensure that it is complete and correct. If you have cited papers that have been retracted, please include the rationale for doing so in the manuscript text, or remove these references and replace them with relevant current references. Any changes to the reference list should be mentioned in the rebuttal letter that accompanies your revised manuscript. If you need to cite a retracted article, indicate the article’s retracted status in the References list and also include a citation and full reference for the retraction notice.

Additional Editor Comments: Add any references that the reviewer has indicated]

Response: Thank you for your valuable feedback. We have thoroughly checked all references and we didn’t get retracted paper. We added reference number 6 to address the impact of spirituality on chronic illness patients that raised by the reviewer. We highlighted the change within the track change and you can also see from clean copy of revised version.

---

## [Editor Report · Decision Letter 2]

28 Feb 2022

Nurse’s spiritual care competence in Ethiopia: a multicenter cross-sectional study

We’re pleased to inform you that your manuscript has been judged scientifically suitable for publication and will be formally accepted for publication once it meets all outstanding technical requirements.

Kind regards,

Luigi Lavorgna

Academic Editor

PLOS ONE
---

## [Editor Report · Acceptance letter]

2 Mar 2022

PONE-D-21-35932R2 

Nurse’s spiritual care competence in Ethiopia: a multicenter cross-sectional study  

Dear Dr. Seid:

I'm pleased to inform you that your manuscript has been deemed suitable for publication in PLOS ONE. Congratulations! Your manuscript is now with our production department. 

Kind regards, 

on behalf of

Dr. Luigi Lavorgna 

Academic Editor

PLOS ONE